Epigenetic considerations in aquaculture

Gavery Mackenzie R.
Roberts Steven B. sr320@u.washington.edu
School of Aquatic & Fishery Sciences, University of Washington , Seattle , WA , USA
Rahman Mohammad Shamsur
Electronic publication date: 2017 Dec 7
Publication date: 2017
Volume: 5
Electronic Location ID: e4147
Received 2017 Sep 1; Accepted 2017 Nov 17
Copyright: ©2017 Gavery and Roberts
Copyright year: 2017
Copyright holder: Gavery and Roberts
License: This is an open access article distributed under the terms of the Creative Commons Attribution License, which permits unrestricted use, distribution, reproduction and adaptation in any medium and for any purpose provided that it is properly attributed. For attribution, the original author(s), title, publication source (PeerJ) and either DOI or URL of the article must be cited.
License URL: https://creativecommons.org/licenses/by/4.0/

Keywords: Aquaculture, Epigenetics, Shellfish, DNA methylation, Histone modifications, Non-coding RNA, Finfish, Seafood

Funding: National Science Foundation (NSF) 1634167 NOAA, National Marine Fisheries Service This work was supported by the National Science Foundation (NSF) under grant number 1634167 (Steven B. Roberts) and NOAA, National Marine Fisheries Service (Mackenzie R. Gavery). There was no additional external funding received for this study. The funders had no role in study design, data collection and analysis, decision to publish, or preparation of the manuscript.

==============================
Epigenetics has attracted considerable attention with respect to its potential value in many areas of agricultural production, particularly under conditions where the environment can be manipulated or natural variation exists. Here we introduce key concepts and definitions of epigenetic mechanisms, including DNA methylation, histone modifications and non-coding RNA, review the current understanding of epigenetics in both fish and shellfish, and propose key areas of aquaculture where epigenetics could be applied. The first key area is environmental manipulation, where the intention is to induce an ‘epigenetic memory’ either within or between generations to produce a desired phenotype. The second key area is epigenetic selection, which, alone or combined with genetic selection, may increase the reliability of producing animals with desired phenotypes. Based on aspects of life history and husbandry practices in aquaculture species, the application of epigenetic knowledge could significantly affect the productivity and sustainability of aquaculture practices. Conversely, clarifying the role of epigenetic mechanisms in aquaculture species may upend traditional assumptions about selection practices. Ultimately, there are still many unanswered questions regarding how epigenetic mechanisms might be leveraged in aquaculture.

Introduction

Maintaining and improving aquaculture production requires an understanding of genetic and physiological mechanisms that control desired traits. Elucidation of these mechanisms has led to the development of pioneering biotechnological methods that have important applications. For example, molecular markers are used in broodstock selection and transcriptomic studies have been used to improve environmental conditions to decrease physiological stress in animals. Recently, interest in epigenetics within the agricultural community has surged as it has become more clear that epigenetic mechanisms can provide a measurable link between environment and phenotype.

Epigenetics refers to processes that result in heritable alterations in gene activity without manipulating the underlying DNA sequence (Jablonka & Lamb, 2002). Epigenetic mechanisms (or ‘marks’), including DNA methylation, histone modifications and non-coding RNA activity, influence gene expression primarily through the local modification of chromatin. Unlike DNA, epigenetic marks can be directly influenced by the environment, and therefore have been shown to be important mediators of phenotypic responses to environmental signals. For example, in mammals nutrition (Weaver et al., 2004), exposure to toxins (Dolinoy et al., 2006), and photoperiod (Azzi et al., 2014) have all been associated with changes in DNA methylation and concomitant changes in phenotype. DNA methylation patterns in fish show a similar sensitivity to the environment (Wang et al., 2009; Strömqvist, Tooke & Brunström, 2010; Campos et al., 2013; Artemov et al., 2017). While many environmentally-induced epigenetic changes are transient, some may persist over the course of an organism’s lifetime (Weaver et al., 2004; Dolinoy et al., 2006; Heijmans et al., 2008). Evidence of transgenerationally-inherited epigenetic changes has been reported in vertebrates (Guerrero-Bosagna et al., 2010; Manikkam et al., 2012; Rodgers et al., 2015; Knecht et al., 2017), invertebrates (Rechavi et al., 2014; Klosin et al., 2017) and plants (Hauser et al., 2011). Thus, it is important to understand the nature and function of these mechanisms and their influence on phenotype in fish and shellfish.

Interest in epigenetics has been gaining ground in agricultural science for crops (Ong-Abdullah et al., 2015; Álvarez Venegas & De-la Peña, 2016) and, more recently, livestock (Goddard & Whitelaw, 2014; González-Recio, Toro & Bach, 2015), but less is known about epigenetic mechanisms in economically valuable aquaculture species. Since most aquaculture operations exist in open or natural conditions that are subject to changes in the environment, it is important to consider the potential role of epigenetics, particularly now that tools are available to study these important phenomena. Recent studies in species ranging from salmonids to oysters and mussels have provided the first evidence that epigenetic mechanisms are associated with commercially important traits in aquaculture species. In sea bass and half-smooth tongue sole, temperature-induced sex-determination has been associated with changes in DNA methylation (Navarro-Martín et al., 2011; Shao et al., 2014). In salmonids, there is some evidence that changes in DNA methylation are associated with variation in life-history phenotypes including early male maturation (Morán & Pérez-Figueroa, 2011), smoltification (Morán et al., 2013), anadromy (Baerwald et al., 2016) and growth potential (Burgerhout et al., 2017). Recent studies in European sea bass and rainbow trout examined the role of epigenetics in mediating phenotypic responses to various aspects of diet (Marandel et al., 2016; Terova et al., 2016; Panserat et al., 2017). In Pacific oysters, the role of epigenetics in mediating the effects of temperature on oyster physiology has been investigated (Fellous, Favrel & Riviere, 2015). Recent efforts have also aimed at understanding the role epigenetics may play in other important phenomena in aquaculture, including sex control via ploidy manipulation (Covelo-Soto et al., 2015; Jiang et al., 2016; Zhou et al., 2016), the effects of inbreeding (Venney, Johansson & Heath, 2016) and adaptation to captivity (Le Luyer et al., 2017).

This review will introduce key concepts and definitions of epigenetic mechanisms, briefly review the literature as it pertains to the nascent field of epigenetics in aquatic species and highlight key aspects of aquaculture that could benefit from a deeper understanding of the role of epigenetics. Several excellent reviews of epigenetics, primarily DNA methylation, and various aspects of finfish aquaculture (e.g., Li & Leatherland, 2013; Moghadam, Mørkøre & Robinson, 2015; Labbé, Robles & Herraez, 2017) have recently been published, and these will be highlighted where appropriate.

Survey methodology

The authors have been actively involved in epigenetic research over the past decade, primarily in shellfish, and more recently in finfish (Gavery). As part of this research, we have performed extensive literature searches and literature reviews in numerous venues including, but not limited to, university library catalogs, Web of Science, Google Scholar, Scopus, and general web searches. In addition, we routinely attend local and international workshops and conferences focused on epigenetics in shellfish and fish, where we interact with colleagues who are actively engaged in similar research. Platforms such as Twitter have also been useful for the discovery of new research and associated manuscripts.

What is epigenetics?

The following section will briefly describe specific epigenetic marks and review where we stand in terms of understanding (or not understanding) the relationship between epigenetics, the environment and phenotype in aquaculture species.

DNA methylation

DNA methylation refers to the enzymatic addition of a methyl group to a cytosine residue in DNA, which occurs almost exclusively at CpG dinucleotides (i.e., a cytosine located 5′ of a guanine) in animals. The enzymatic machinery supporting DNA methylation includes a family of DNA methyltransferases (DNMTs), including the maintenance methyltransferase DNMT1 (responsible for copying pre-existing DNA methylation patterns to the new strand during mitosis) and the de novo methyltransferases DNMT3A/3B. DNA methylation is known to be repressive when located in promoters of genes through associations with other DNA-binding proteins or through the physical blocking of transcription factors (Bell & Felsenfeld, 2000). However, DNA methylation in gene bodies is associated with high levels of expression (Jones, 1999). Therefore, although DNA methylation is typically associated with silencing, its regulatory role is specific to the genomic context. In mammals, DNA methylation plays important roles in providing genomic stability through the repression of transposable elements (TEs) (Maloisel & Rossignol, 1998), genomic imprinting (Bell & Felsenfeld, 2000), and dosage compensation (Csankovszki, Nagy & Jaenisch, 2001). DNA methylation is also important for cell-type differentiation and embryonic development (Li, Bestor & Jaenisch, 1992). DNA methylation is the most well-studied epigenetic mechanism, and most studies have been done in plants and mammals, in which DNA methylation has been shown to be sensitive to external factors including nutrition (Weaver et al., 2004), exposure to toxins (Dolinoy et al., 2006), and photoperiod (Azzi et al., 2014). Importantly, the meiotic transmission of DNA methylation patterns, and thus the opportunity for transgenerational epigenetic inheritance through DNA methylation, is rare in mammals, which undergo extensive DNA methylation reprogramming in the early embryo stage (Daxinger & Whitelaw, 2012). Transgenerational epigenetic inheritance is more common in plants, which do not exhibit extensive resetting of DNA methylation between generations (reviewed by Hauser et al., 2011). As discussed below, it is still unclear if and to what extent DNA methylation resetting occurs in fish and shellfish.

Histone variants and post-translational modifications

Chromatin is a dynamic structure that supports both packaging of the genome into the nucleus and the regulation of genes and other genomic regions via changes in DNA accessibility (Cheung, Allis & Sassone-Corsi, 2000). The basic repeating structure of chromatin is the nucleosome, which consists of DNA wrapped around an octamer of four core histone proteins (H2A, H2B, H3 and H4). Higher-order chromatin structure is established via the incorporation of linker histones (H1) between nucleosomes. Chromatin structure can be modified to either enhance or repress transcription through the incorporation of histone variants and the post-translational modification of histones (Berger, 2007). These modified chromatin states can be inherited both mitotically and meiotically, and thus may convey epigenetic information (Henikoff & Smith, 2015). Histone variants have specialized functions and can be incorporated into nucleosomes in a replication-independent manner (Henikoff & Smith, 2015). Although histone variants have been less-studied than post-translational modifications, they can play an important role in mediating both short- and long-term responses to environmental cues (Talbert & Henikoff, 2014). Both canonical and variant histones can be post-translationally modified, primarily at their N-terminal tails, which alters the degree of chromatin compaction resulting in either euchromatin (referring to open chromatin that is accessible to transcription factors, RNA polymerase II (Pol II) and other DNA-binding proteins that support gene expression) or heterochromatin (referring to tightly packed DNA associated with transcriptional silencing). These states are dependent on the type (e.g., acetylation, methylation, phosphorylation, ubiquitylation) and location (e.g., various lysine or arginine residues) of the modification (see review by Lawrence, Daujat & Schneider, 2016 for a complete list of modifications). These modifications are enabled by various families of enzymes, including histone acetylases (HATs), histone deacetylases (HDACs), histone methyltransferases (e.g., HMT) and histone demethylases (e.g., Jumonji and Lys-specific demethylase). Post-translational modifications are important for the regulation of gene activity, but also play roles in DNA repair, replication, and cell fate/determination (see reviews by Eberharter & Becker, 2002; Martin & Zhang, 2005; Lawrence, Daujat & Schneider, 2016). The enzymatic machinery responsible for these modifications is highly regulated during embryonic development (Lin & Dent, 2006), and, like DNA methylation, can be altered by various environmental conditions (Chinnusamy & Zhu, 2009). Less is known about the mechanisms that underlie the mitotic and meiotic persistence of histone modifications. Interestingly, it has been shown in both mammals and zebrafish that certain modified histones are non-randomly retained during spermatogenesis when most of these proteins are replaced by protamines, suggesting that these marks may play a role in transferring epigenetic information to the embryo (Brykczynska et al., 2010; Wu, Zhang & Cairns, 2011).

Non-coding RNA

Although a large majority of the genome is transcribed, only a small portion of these transcripts actually code for proteins. The remaining non-coding transcripts, originally regarded as ‘junk’, are now recognized to play a role in modulating gene expression, and are categorized broadly as non-coding RNA (ncRNA). There are two major classes of ncRNA: long ncRNA (>200 nt) and small ncRNA (<200 nt), the latter of which includes micro RNA (miRNA), short interfering RNA (siRNA), and PIWI-interacting RNA (piRNA). Small ncRNAs are highly conserved and their major mechanism of action is to inhibit protein synthesis by blocking or degrading primary transcripts (see review by Castel & Martienssen, 2013). In contrast, long ncRNAs (lncRNA) are less conserved and have complex mechanisms of action that may work either in cis or trans (see review by Wang & Chang, 2011). Non-coding RNAs have important functions in gene expression and have been demonstrated to be important regulators of genome stability, environmental plasticity and embryonic development (Mercer, Dinger & Mattick, 2009; Bizuayehu & Babiak, 2014). Generally, ncRNA molecules are considered to be ‘epigenetic’ in the traditional sense because they interact with other epigenetic mechanisms such as DNA methylation and histone modifications to silence or activate various parts of the genome (Peschansky & Wahlestedt, 2014). There is also evidence that ncRNA, particularly miRNA, plays a role in the transmission of environmental information from the male parent (via sperm) to offspring in mammals (Gapp et al., 2014; Rodgers et al., 2015). Evidence of transgenerational inheritance through miRNA has also been observed in the invertebrate model C. elegans in response to starvation (Rechavi et al., 2014) and temperature (Klosin et al., 2017) cues.

Taxa specific patterns

Epigenetic mechanisms, and particularly DNA methylation, have recently been the focus of numerous studies in both fish and shellfish. However, most of what we know about epigenetics in animals comes from studies done in mammals and care should be taken when generalizing results in mammals to fish and shellfish. Although there are certainly similarities (e.g., DNA methylation patterns are very similar across all vertebrates), there are also important differences (e.g., invertebrate DNA methylation patterns are very different from those in vertebrates). This section will focus on foundational information about epigenetic marks in fish and shellfish, and highlight both significant gaps in our understanding and differences from well-studied mammalian systems.

DNA methylation in fish and shellfish

DNA methylation is the most well-studied epigenetic mark among fish and shellfish. Both fish and shellfish have genes that encode the basic methylation machinery (e.g., DNMTs and MBDs) and DNA methylation is present in all species examined to date. However, there are striking differences in DNA methylation patterns between vertebrates and invertebrates, as well as significant unknowns in terms of the resetting of DNA methylation in both fish and shellfish, which will be described below.

Considerable work has been done on understanding patterns and functions of DNA methylation in model fish species such as zebrafish and medaka, and there is also a growing body of information on DNA methylation in non-model species (e.g., Metzger & Schulte, 2016 reviewed the current state of knowledge of DNA methylation patterns and functions in marine fish). Generally, DNA methylation patterns are similar across all vertebrates which exhibit a ‘global’ DNA methylation pattern, which means that most CpGs are methylated with the exception of regions of DNA with a high CpG content (referred to as CpG islands). However, global DNA methylation levels in fish are higher than those in mammals, though the significance of this difference remains unclear (Jabbari et al., 1997; Zhang, Hoshida & Sadler, 2016). The function of DNA methylation also appears to be similar across vertebrates, with the exception that its role in parental imprinting is likely unique to mammals (Barlow & Bartolomei, 2014). One outstanding question concerns the extent of DNA methylation resetting in fish. While mammals undergo extensive DNA methylation reprogramming in the early embryo (Daxinger & Whitelaw, 2012), the extent of DNA methylation reprogramming in fish is unclear (Jiang et al., 2013; Potok et al., 2013). A recent study, which is discussed in more detail in the following section, showed clear evidence of transgenerational inheritance of environmentally-induced DNA methylation patterns in a fish, suggesting that at least some of the genome escapes putative resetting between generations (Shao et al., 2014). There is a need for more detailed studies on the extent of DNA methylation resetting in fish, particularly in species used in aquaculture. In addition, more studies should examine the potential meiotic inheritance of environmentally-induced epigenetic changes.

Invertebrate DNA methylation patterns are strikingly different from those in vertebrates. Whereas vertebrates exhibit a global pattern of DNA methylation, invertebrates show a ‘mosaic’ pattern, with stretches of methylated DNA punctuating regions of unmethylated DNA (Tweedie et al., 1997; Simmen et al., 1999). We previously examined DNA methylation throughout the entire genome in the Pacific oyster; 15% of CpGs in somatic tissue were methylated, whereas 60–70% of CpGs are methylated in mammals (Gavery & Roberts, 2013). In oysters, as in other invertebrates, the methylated fraction tends to consist of gene bodies, while other genomic regions exhibit less methylation. Unlike vertebrate species, transposable elements in oysters and other invertebrate species show surprisingly little methylation (Simmen et al., 1999; Feng et al., 2010; Zemach et al., 2010). Despite these differences, DNA methylation does appear to be associated with gene regulation in shellfish. In the Pacific oyster, high levels of methylation in gene bodies are associated with high levels of expression (Gavery & Roberts, 2013; Olson & Roberts, 2014). Interestingly, genes in oysters with limited methylation show variable exon-specific expression across tissue types, indicating that hypomethylation allows increased plasticity (Gavery & Roberts, 2013). DNA methylation patterns in the Pacific oyster are dynamic across developmental stages, and correlations between DNA methylation and gene expression patterns suggest that DNA methylation plays a role in gene regulation during development (Riviere et al., 2017). While more studies are needed to quantify the functional relationship between DNA methylation and gene expression, there are significant implications for improving resilience in shellfish—particularly if DNA methylation patterns are heritable. While few studies have examined the heritability of DNA methylation patterns in shellfish, a small study that focused on methylation states in parents and larvae found a significant clustering of methylation patterns within families, indicating that methylation patterns differ significantly depending on the male parent (Olson & Roberts, 2015). More recently, Rondon et al. (2017) showed that parental exposure to an herbicide influences progeny DNA methylation patterns in oysters.

Histone variants and post-translational modifications in fish and shellfish

Histone variants and post-translational modifications have generally been studied in model fish species, such as zebrafish. Some histone variants show a conserved function between lower vertebrates and mammals. For example, histone variant macroH2A has been shown to play an important role in development in the zebrafish (Buschbeck et al., 2009). However, while some histone variants are ‘universal’ and can be found in most eukaryotes (Talbert & Henikoff, 2010), other histone variants appear to be unique to fish (Wu et al., 2009).

Post-translational modifications of histones and their dynamics have also been studied in zebrafish, and the evidence indicates that modifications are conserved among vertebrates. The functional analysis of histone acetylation in zebrafish confirms that it plays a role in embryogenesis (Vastenhouw & Schier, 2012) and in tissue regeneration (Stewart, Tsun & Izpisu Belmonte, 2009). While studies of histone modifications in non-model fish are rare, recent studies in rainbow trout and European sea bass indicate that diet influences bulk histone modification levels and can regulate the expression of associated enzymes (Marandel et al., 2016; Terova et al., 2016; Panserat et al., 2017). With regard to meiotic inheritance, zebrafish show multivalent modified histone retention in sperm, similar to mammals (Wu, Zhang & Cairns, 2011).

Histone variants have been characterized in various bivalve species (e.g., González-Romero et al., 2009; González-Romero et al., 2012) and a recent study on the responses of Eastern oysters to harmful algal blooms reported that the histone variant H2A.X is post-translationally modified in response to algal toxins (Gonzalez-Romero et al., 2017). While post-translational modifications of histones are not well-studied in shellfish, Fellous et al. (2014) identified homologs of Jumonji histone demethylase genes (Jmj) in Pacific oysters that, as in vertebrates, were regulated during embryonic development. A subsequent study showed that both bulk histone methylation levels and the expression of histone demethylases responded to temperature during development, suggesting that histone modifications play a role in mediating the physiological responses of oysters to temperature (Fellous, Favrel & Riviere, 2015).

Histones are not replaced by protamines in bivalve sperm as they are in mammals (Eirín-López & Ausió, 2009). Rather, depending on the species, canonical histones are replaced by various sperm nuclear basic proteins that can be classified as either protamine-like-type, histone-type or protamine-type (Ausio, 1986; Eirín-López & Ausió, 2009). Further studies will be needed to determine the extent to which canonical histones and variants are retained in bivalve sperm, and this avenue of research could greatly contribute to a greater understanding of the potential transmission of epigenetic information from parent to offspring in bivalves. Recent work on the details of experimental approaches and workflows for the study of chromatin-associated proteins in bivalves (Rivera-Casas et al., 2017) should facilitate this research.

Non-coding RNA in fish and shellfish

Most studies on non-coding RNAs in fish and shellfish, including important aquaculture species (e.g., Atlantic salmon (Andreassen, Worren & Høyheim, 2013; Bekaert et al., 2013) and rainbow trout (Juanchich et al., 2016)), have focused on miRNAs. There are several examples of examining miRNAs in a physiological context including: immune function (e.g., Andreassen et al., 2017) and embryonic development (e.g., Ma et al., 2012; Bizuayehu et al., 2015). There is less information available about other types of small ncRNA, except in zebrafish where, for example, piRNAs have been shown to silence transposable elements in gametes, and thus functions similarly to that in mammals (Houwing et al., 2007). Recently, there have been several descriptions of long non-coding RNAs in salmonids, including associations between lncRNA expression and disease in both Atlantic salmon and rainbow trout (Boltaña et al., 2016; Paneru et al., 2016; Valenzuela-Miranda & Gallardo-Escárate, 2016).

While there have been few studies on non-coding RNAs in shellfish, generally, miRNAs and their biogenesis are highly conserved over evolutionary scales (Wheeler et al., 2009). Indeed, genes for miRNA biogenesis have been detected in available bivalve genomes (Rosani, Pallavicini & Venier, 2016). With respect to long non-coding RNAs, researchers have reported an association between lncRNAs and larval development in the Pacific oyster (Yu, Zhao & Li, 2016).

Potential aquaculture applications

Environmental manipulation

Given what we know about environmental influences on epigenetic mechanisms in fish and shellfish and the relationships between these mechanisms and phenotype, a potentially fruitful application of epigenetics to aquaculture could involve environmental manipulation. The possibility of generating environmentally-driven phenotypes mediated through epigenetic mechanisms should be considered for two stages in the aquaculture life-cycle in which the animals are particularly sensitive: during larval development and during broodstock holding/conditioning (Fig. 1).

Figure 1 Schematic diagram of the aquaculture life-cycle highlighting key areas where epigenetics could be applied to improve productivity and efficiency.

Epigenetic selection (red text) could be used, alone or in combination with genetic selection, to identify individuals with desired traits. Environmental manipulation (blue text) refers to generating environmentally-driven phenotypes mediated through epigenetic mechanisms. Two life stages which may be particularly sensitive to generating within- or between-generation ‘epigenetic memories’ are larvae and broodstock, respectively.

The notion of developmental programming suggests that environmental conditions experienced in early life influence the phenotype later in life, and has gained momentum in human research (e.g., Gluckman et al., 2008). Thus, developmental programming offers a memory of the environment that could be beneficial in controlled aquaculture settings. However, in some cases, embryos and juveniles are not raised in the same environmental conditions as the adults. For example, hatchery-reared salmon or bivalves experience very different conditions during their larval and adult stages. The identification of sensitive periods for inducing an environmental memory could offer a “programming window” that could be leveraged in husbandry practices. Several lines of evidence support the presence of developmental programming in fish (for an excellent review see Jonsson & Jonsson, 2014). Traits that have been associated with early environmental conditions include metabolism, growth, sex determination, fecundity, and behavior (Jonsson & Jonsson, 2014). Within-generation environmental memory has also been described in shellfish. For example, early exposure of Olympia oyster larvae to ocean acidification has been shown to influence juvenile traits (Hettinger et al., 2012).

Two examples of developmental programming in aquaculture, where the epigenetic mechanisms were described, involve sex determination in fish. In European sea bass (Dicentrarchus labrax), exposure to high temperature in early development was associated with a higher proportion of phenotypic males (Navarro-Martín et al., 2009). Navarro-Martín et al. (2011) found that this early exposure to high temperature was associated with increased DNA methylation in the promoter of the aromatase gene (cyp19a1a) in adults. Furthermore, they showed that in vitro methylation of the aromatase promoter was sufficient to suppress transcription of the gene. More recently, the commercially important half-smooth tongue sole (Cynoglossus semilaevis) was used as a model to investigate the role of epigenetic regulation in environmental sex determination (Shao et al., 2014). Using genome-wide DNA methylation profiling, the authors showed that pseudomales (generated by exposing genetic females to high temperature during a sensitive developmental window) exhibit methylation patterns in testes consistent with genetic males, both of which differ from the ovarian methylome of normal females. Excitingly, it was reported that both the pseudomale phenotype and the testes-specific methylation patterns are inherited by F1 pseudomale offspring generated by crosses between pseudomales and normal females, suggesting the transgenerational epigenetic inheritance of environmentally-induced sex reversal in this species (Shao et al., 2014). The ability to control sex in fish broodstock is certainly a priority for aquaculture and these studies shed light on the epigenetic mechanisms that could be leveraged in future work. More studies will be needed to determine the extent to which the relevant mechanisms are conserved across species that exhibit environmental sex determination.

Nutrition and feeding are important aspects of aquaculture production; and understanding how early-life nutritional conditions influence key phenotypic traits later in life is important to consider. In mammals, the nutritional status of the mother can predispose offspring to adult-onset metabolic disease and mounting evidence suggests that epigenetic mechanisms are involved (reviewed by Vickers, 2014). In fish, rainbow trout fry fed a plant-based diet for 3 weeks starting when they were swim-up fry showed higher growth rates, feed intakes, and feed efficiencies when challenged again with a plant-based diet after 7 months of grow-out on a fishmeal/fish oil diet (Geurden et al., 2013). Interestingly, in a follow-up study, transcriptomic analyses suggested that epigenetic mechanisms may be involved in this response (Balasubramanian et al., 2016). In addition, a study on vitamin supplementation at first feeding in rainbow trout identified changes in global methylation and histone modification 7 months after the supplementation was discontinued, despite an apparent lack of phenotypic responses (Panserat et al., 2017). These studies provide the first link between early-environmental exposure and epigenetic mechanisms in aquaculture species.

In addition to developmental programming, broodstock holding/conditioning is also an important area to consider for the potential transmission of environmentally-induced epigenetic information between parents and their offspring. This type of non-genetic transmission is frequently referred to as ‘transgenerational plasticity’ (Salinas et al., 2013) and can refer both to maternal provisioning (e.g., bivalve embryos can be influenced by the type of diet fed to broodstock during conditioning (Utting & Millican, 1997)) and epigenetic inheritance. Importantly, epigenetic transmission has the potential to be transmitted not only from the maternal side, but also from the paternal side, which may be more important than previously thought (Rodgers et al., 2015; reviewed by Soubry et al., 2014). There has been growing interest in the transgenerational plasticity of fish, particularly as it relates to predicting responses to climate change (reviewed by Munday, 2014). Similar research questions are also being addressed in shellfish. Adult Manila clams exposed to low pH during gonadal maturation have faster-growing offspring compared to controls (Zhao et al., 2017). In the Sydney rock oyster, larvae produced by parents incubated under low-pH conditions are larger and develop faster in low-pH conditions and also have higher fitness as adults (Parker et al., 2012; Parker et al., 2015). In addition to water chemistry, disease is a significant concern in shellfish aquaculture. There is increasing evidence that prior exposure to an immune challenge can increase the immune response later in life and that this environmental memory can be transmitted to offspring. Green et al. (2016) demonstrated that offspring of Pacific oyster parents treated with poly(I:C) possess enhanced protection against Ostreid herpesvirus type I infection.

The mechanism(s) responsible for providing this memory of the environment in cultured species are not fully understood. While such an understanding is, arguably, not required to improve aquaculture production, we would suggest that elucidation of the epigenetic mechanisms involved could increase the degree and breadth of ongoing improvements.

Epigenetic selection

It is possible that epigenetic markers could be integrated into broodstock selection (Fig. 1). The concept of epigenetic selection has gone from theory to practice in one important agriculture commodity, oil palms, where it has been shown that it is possible to epigenetically select for a critical trait: oil content (Ong-Abdullah et al., 2015). While there is much more we need to learn with regard to desired phenotypes and epialleles, work such as this demonstrates that it can be useful to consider epigenetics in association studies. Furthermore, Patel et al. (2013) showed in a clinical study that the use of both genetic (SNP) and epigenetic (DNA methylation) markers in genome-wide association studies improved associations with a phenotype (i.e., diabetes). The influence of epigenetics, specifically DNA methylation, on the ability to estimate breeding values for quantitative traits has recently been considered for finfish aquaculture (see the review by Moghadam, Mørkøre & Robinson, 2015).

Epigenetics could also make genetic selection more challenging. Many organisms have the potential to generate new genetic variation in response to stressful conditions through the modulation of epigenetic marks associated with transposable elements (TEs) (Dowen et al., 2012; Yu et al., 2013; reviewed by Rey et al., 2016). Transposable elements, or “jumping genes”, are regions of repetitive DNA that can move and amplify their copy number in the host genome. In the model plant Arabidopsis, the genomic response to bacterial challenge is a global reduction of DNA methylation and the reactivation of previously silent TEs associated with defense genes (Yu et al., 2013). It is interesting to consider that in invertebrates, and specifically in Pacific oysters, TEs are not preferentially methylated, though this does not preclude silencing via other epigenetic mechanisms (Gavery & Roberts, 2013; Olson & Roberts, 2015). It has been hypothesized that the lack of TE silencing by DNA methylation may indicate pressure to generate and maintain genetic diversity in a species that inhabits heterogeneous environments (Gavery & Roberts, 2014). This means that, in theory, if the culture conditions become stressful, shellfish could respond by modulating transposable element expression to create new genetic variation (Rey et al., 2016), thereby having the unintended consequence of “erasing” phenotypic gains made through selective breeding.

Conclusions

Epigenetics has the potential to change the way we think about how a phenotype is generated and maintained. Through a greater understanding of DNA methylation, histone modifications and ncRNAs, we can functionally annotate genomes, better predict phenotypic outcomes of early environmental exposures, and possibly select based on epigenetic markers. With careful experimental design and special considerations for epigenetic differences between taxa (see Lea et al., 2017), the aquaculture community is primed to begin to integrate epigenetics into husbandry practices. The concepts and ideas of epigenetics provide an attractive lens through which to consider the manipulation of traits through environmental memory or the selection of beneficial traits based on epigenetic markers. It is also important to consider that epigenetics may also function to disrupt predictable, robust phenotypes through the creation of new, unexpected variation.

The authors would like to thank Guillaume Rivière and the anonymous reviewer for their helpful and constructive comments that greatly contributed to improving the final version of the paper.

Additional Information and Declarations

Competing Interests

Author Contributions

Data Availability

The authors declare there are no competing interests.

Mackenzie R. Gavery and Steven B. Roberts wrote the paper, prepared figures and/or tables, reviewed drafts of the paper.

The following information was supplied regarding data availability:

The research in this article did not generate any data or code. This is a literature review.

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
