# Peer review of "Epigenetic considerations in aquaculture"

_PeerJ, doi:10.7717/peerj.4147_

## Round 0.1 · original submission · Minor Revisions

Please carefully address all the suggestions of reviewers' and resubmit the manuscript

·

Basic reporting

The importance of epigenetics in different fields of biology including fundamental molecular biology, evo-devo problematic and more applied ecological contexts has led to a tremendous increase in the literature, urging for appropriate syntheses of the recent knowledge and perspective in the related fields. Here the authors propose a review about the 'epigenetic considerations in aquaculture' that suits the readership of the journal in my opinion. The work is divided into two parts, with the first dedicated to a short but comprehensive description of epigenetic mechanisms in the different taxa that are considered in the manuscript, which are the most important for aquaculture, fish and shellfish. Then the authors provide a clear perspective of the broad interests of application of epigenetic knowledge for aquaculture present practices and future developments.
The manuscript is well written and the quality of the writing was really appreciated. I also congratulate the authors for bringing together data from vertebrate and invertebrate groups. This constitutes to me the strength of the manuscript and its most important contribution especially when compared to a review by Moghadam et al. in 2015 on the same topic but that 'only' considered fish.
Nevertheless there are a few substantial comments that should be addressed before acceptance (see other sections below), mostly clarification of apparent contradiction and inclusion of recently published findings.

Experimental design

Why did the authors choose to use Google scholar to browse the literature? There are known issues with the quality/relevance/extensiveness of the results provided by this service when compared to 'classical' scientific literature databases (Lewandowsky et al. Online Information Review 2010,Levine & Gil 2009). I believe this might be due to the lack of such a unique database covering the width of the reviewed topic, especially for aquaculture journals. Please provide justification in the text.

Validity of the findings

Although the literature reviewed is mostly cited in a relevant, exhaustive and comprehensive manner, there are some places where the authors need to provide additional data, details or explanations of their interpretation of the cited references at the cost of what seemed to me incomplete or contradictive review. There are two places in particular:

- Lines 201-203: ' The function of DNA methylation also appears to be similar across vertebrates with the exception of a role in genomic imprinting which is unique to mammals (Potok et al. 2013)'. In this paper, Potok et al. considered 'imprinting ' in the sense of McGowan and Martin, ie. ' imprinting results in an inability to reproduce parthenogenetically because a genetic contribution from both parents is necessary to successfully complete development.' (McGowan and Martin, 1997). However, imprinting is a broader phenomenon which is mostly considered as 'the non-equivalent contribution of the parental genomes to the embryonic genome', or 'the preferential expression of a parental epiallelle over the other'. The existence of such different contribution of parental nuclear genomes, ie imprinting, has been demonstrated in fish and is described by the authors lines 325-332 ' Using genome-wide DNA methylation profiling, authors showed that pseudomales (generated by exposing genetic females to high temperature during a sensitive developmental window) exhibit methylation patterns consistent with genetic males, both of which differ from the methylome of normal females. Excitingly, it was reported that global methylation patterns are inherited by F1 pseudomale offspring generated by crosses between pseudomales and normal females, suggesting transgenerational epigenetic inheritance of environmentally-induced sex reversal in this species (Shao et al 2014).'. Those two claims contradict one another to me. Please clarify.

- Lines 225-226 : remove '(and putative promoter regions)', which is debated and not assumable regarding the present knowledge.

-Line 336-350: Please include discussion on nutritional influence on phenotype in bivalves (broodstock conditionning....).

- The literature about DNA methylation in marine organisms has significantly increased very recently, and brings data that should be included here. importantly, please consider Knecht et al Toxicol Appl Pharmacol 2017, Burgerhout et al PloS one 2017 and Artemov et al Mol Biol Evol 2017 for the fish; Riviere et al PLoS Genetics 2017 and Gonzalez-Romero Aquat Toxicol 2017 for the oyster.

An important issue in aquaculture is the control of the reproduction cycle of exploited species. The manuscript deals with sex determination but would gain an additional asset if presenting a short paragraph on this topic on both fish and bivalves. See for example Coveto-Solo et al. Anim Genet 2015, Zhou et al Int J Mol Sci 2016 for fishes, Jiang et al Mar Biotechnol 2015 in the oyster.

The reference list needs format editing for homogeneity.

Reviewer 2 ·

Basic reporting

The present manuscript represents a very interesting effort in bringing together current epigenetic and marine biology knowledge. It is especially interesting the insight this work provides concerning the potential of epigenetic approaches to improve aquaculture practices. Epigenetics is a relatively young discipline, nonetheless, it constitutes the basis for most molecular biology research done in model organisms nowadays (e.g., cancer research). Still, this approach lacks development in other fields encompassing critical environmental components such as ecology, marine biology, and fisheries, just to mention some. In that sense, the present work has the potential to make a great contribution to bring epigenetics into the latter, helping improve current management, conservation, and restoration strategies.
I found the present manuscript well organized in essence, with most of the relevant literature cited as well as many gaps in knowledge addressed. That being said, I believe the work needs substantial improvements to live up to its title. If this work is meant to be a foundational reference (as mentioned by the authors at some point), it requires further elaboration on critical points, as well as additional references and contents completing the information provided.
Overall, I think this work is promising, however, I am recommending major revision before further considering it for publication in Peer J. I believe authors can improve this work substantially and I hope the comments below might help them do that in the best way possible.

Experimental design

N/A

Validity of the findings

N/A

Additional comments

Abstract: Current narrative is a bit cloudy with some awkward sentences “… has dramatically increased to where we are now able to characterize …”. Overall, my suggestion is to make abstract more descriptive of the key concepts developed throughout the manuscript.
Line 54: refeernces for Wang et al 2009, and for Wu et al. 2011, are duplicated in the references section.
Figure 1: Is it correctly represented? Why is there black and grey text? Why is epigenetics box intersecting with genetics? Why not with environment? Where are the references supporting the different markers used? These should be indicated in the figure. Please, provide a more coherent and detailed explanation of this figure.
Line 89: Survey methodology is not really appropriate as a section in the manuscript. This looks like some sort of justification in case any relevant references are missing (blame Google!). It seems a bit narrow the fact that this revision can just be based on a google search, rather than on author’s genuine knowledge and passion for the topic through the years.
Line 122: what about plants? Since general background is given here, it is relevant to mention that trans-generational transmission of DNA methylation has been observed in plants. Please elaborate.
Line 127: Histone modifications are explained here but the epigenetic role of histone variants is completely neglected? What about their role (and the role of their PTM modifications) during epigenetic regulation? This must be addressed here.
Line 133: degree in which DNA wrapped around would be probably better explained in terms of electrostatic affinity between DNA and histones?
Line 132: core histones are modified at N terminal tails, but linker histones are modified at C terminal tail, it would be better to say that PTMs occur mostly at N terminal tails.
Line 136: “These states are dependent on the type (i.e. …) since not all modification types are mentioned here, it is probably better to use “e.g., …”. There are many other less know PTMs in addition to the 4 mentioned (sumoylation, Pro-isomerization, ADP ribosylation, biotinilation …), although this is not a review about PTMs, authors should at least aknowledge that more PTMs exist.
Line 137: different chromatin states not only depend on modifications, they also depend on histone variants, extensively!
Line 139: Lawrence et al 2016 citation lacks details in the references section.
Line 141: the concept of the histone code has been losing momentum during the last few years, not sure if relevant to mention that here. At any rate, it would be interesting to at least dedicate 1 sentence to explain what the histone code hypothesis is about.
Line 144: add “among many others” at end of sentence, there are many other histone-modifying enzymes not mentioned here.
Lines 145-146: references here (except Lawrence) are a bit outdated, I suggest updating this list, some options could include
http://www.readcube.com/articles/10.1038/nrg.2016.59
https://www.ncbi.nlm.nih.gov/pubmed/24614311
Line 147: “modifications IS highly regulated …”
Line 157: “code for proteinS …”
Line 171: Information about trans-generational inheritance of RNAs is lacking, need to include that, see ref: http://www.readcube.com/articles/10.1038/nn.3695
Line 191: I am missing a reference justifying the affirmation made by that sentence.
Line 194: Same, reference at end of sentence will be better.
Line 239: There is a significant amount of knowledge about histone variants in shellfish, (see work by Jose Eirin-Lopez’s lab), since everything epigenetic seems so scarce in shellfish, I believe those are worth mentioning. Similarly, this group has recently published a paper summarizing epigenetic methods for the study of shellfish epigenetics, I’m surprised this is not even mentioned here: https://www.ncbi.nlm.nih.gov/pubmed/28848447
Line 254: Recent work has addressed the role of histone H2A.X phosphorylation on oyster responses to toxins: https://www.ncbi.nlm.nih.gov/pubmed/28315825
Line 255: Bivalves replacing histones by protamines in sperm DO NOT EXIST. Bivalves either replace histones by sperm-specific histones or by protamine-like proteins. The only molluscs with protamines in sperm are cephalopods. I suggest that authors check https://www.ncbi.nlm.nih.gov/pubmed/19708021 for additional information.
Line 278: This section is where this manuscript starts discussing topics never discussed before. This should be the strength of the present work. I encourage authors to revise and further elaborate this section of the manuscript.
Line 285: how does “developmental programming” relates to the concept of hormesis?
Line 352: would it be interesting to incorporate the concept of “assisted evolution” in the case of aquaculture? This has been described for corals, raising a lot of controversy, would it make sense in the present context?

---

## Round 0.2 · accepted · Accept

In various fields, now-a-days, epigenetics are applied; so, why not in aquaculture and fisheries! Thank you for your contribution! hope this article will attract more researchers to do work with epigenetic approaches to improve the quality of aquaculture production!